# Spatio-Temporal Patterns and Consequences of Road Kills: A Review

**DOI:** 10.3390/ani11030799

**Published:** 2021-03-12

**Authors:** Ayrton Gino Humberto Emilio Oddone Aquino, S’phumelele Lucky Nkomo

**Affiliations:** Discipline of Geography, School of Agriculture, Earth & Environmental Sciences, University of KwaZulu-Natal, Durban 4041, South Africa

**Keywords:** animal-vehicle collisions, environmental impacts, habitat fragmentation, road kill, road infrastructure, spatiotemporal patterns

## Abstract

**Simple Summary:**

Road kill continues to be a challenge in the 21st century. Numerous studies have sought to explain the causes and risks of animal-vehicle collisions that result in road kills, and how best to mitigate these events. This review evaluates the relevant literature on road kills, in order to determine how to effectively address them. Identifying methodologies and sources used in previous studies, how mortalities are normally recorded and reported was determined. Previous literature has suggested that spatial proximity, road infrastructure, traffic volume and velocity, driver awareness, landscape, climate and weather conditions, and animal behavior are the primary factors contributing to the spatio-temporal patterns of road kills. Important socioeconomic and environmental impacts of animal-vehicle collisions that result in road kills were also identified. Current mitigation measures for addressing road kills were examined from previous studies; including road management and wildlife crossing structures. Shortcomings to strategies and methodologies for addressing animal-vehicle collisions were subsequently assessed. Thereafter, the paper analysed geospatial technologies that have been utilised inroad kill studies. This review recommends focusing an all road kills in an area, using larger study locations, taking timelier observations, the increased use of citizen science, more research on nighttime driving speeds, and popularising effective road kill apps.

**Abstract:**

The development and expansion of road networks have profoundly impacted the natural landscape and various life forms. Animals are affected by these roads in a myriad of ways, none as devastating as road mortalities. This article reviews the literature on the magnitude, spatiotemporal patterns, factors, and consequences of Animal-Vehicle Collisions (AVCs) and the subsequent road kills. Furthermore, the review paper briefly outlines the relationship between roads and animals in the surrounding landscape and later examines the nature and impacts of AVCs. This article evaluates the statistics on the number of road kills and a critical analysis of the spatiotemporal patterns of these mortalities is also evaluated. Subsequently, the review paper examines current mitigation measures and the challenges impeding their success. The paper then concludes with an evaluation of geospatial tools (GIS) and other technologies used in road kill studies. The relevant findings of this paper are that, (1) factors influencing road kill patterns interact with one another; (2) AVCs have serious environmental, economic and social consequences; (3) road kill mitigation strategies suffer several challenges hindering their success; and (4) specific geospatial tools and other technologies have been utilised in assessing AVC road kill patterns. The review, therefore, recommends including overall road kill clusters of all animals in mortality surveys, increasing the spatial coverage of road kill observations, consistent surveying, sufficient research on nighttime driving distances and speed, utilising citizen science in all road mortality studies and incorporating GIS into all apps used for recording road kills. An increased sufficiency in road kill data coupled with improved technologies can enable more effective mitigation strategies to prevent AVCs.

## 1. Introduction

The development and growth of road networks profoundly impact on the surrounding landscape and various life forms. Increasing population size and economic growth facilitates the demand for transportation infrastructure at all scales [1,2,3]. The expansion of transport infrastructure has contributed to unquestionable economic and social development [4,5]. On the other hand, rapid road infrastructure development has significantly impacted on the natural landscape alterations and degradation globally [2,4,6]. Road infrastructure development with steep and sensitive terrain results in environmental degradation, of which common features are roadside erosion (predominantly gullies), slumping/landslides, and rock fall, amongst others [1,7,8]. In addition to environmental degradation, road infrastructure development also affects animal (wild and domestic) movement. Vehicle collisions with animals persist in being a major challenge [1,2,4,9,10,11,12,13]. Animals have to maneuver through these modified landscapes, and face the risk of a collision with a vehicle on the road [3,9,14]. Hence, the increased expansion of road networks increases the risk of an Animal-Vehicle Collision (AVC) at a given time and place. High risks of AVCs inevitably imply an increased risk of road kills. The people involved in an AVC often experience some form of personal injury or damage to their vehicles, particularly if the vehicle collides with a large animal [10]. In extreme cases, AVCs can result in human fatalities [10,15,16]. Consequently, AVCs are a serious road safety issue. The rationale for focusing on AVCs in the present study, thus, emanates from the need to understand why animal road kill persists to be a common challenge throughout many countries in the 21st century.

Bartonička et al. [1] argue that strategies to combat AVCs have been either poorly implemented or hindered by research gaps. In particular, research in this context has been plagued with inconsistencies in road kill surveying, methodologies, and subsequent findings. For example, many road kill studies are taxonomy-based, focusing on grouping or targeting specific species affected by AVCs, instead of considering all animals in the study area [1,17]. A complete understanding of the nature and impact of animal-vehicle collisions is necessary to resolve the resultant challenges. It is important to note that not every AVC results in the vehicle incurring damages, or the driver and their passengers suffering any injury or fatality. Collisions with butterflies or frogs are unlikely to affect the vehicle or people inside it. Furthermore, some animals are injured, rather than killed, in an AVC [18]. Nevertheless, all road kills result from AVCs. Relevant authorities in road management and safety will usually address AVCs regarding how they affect the people involved. Conservationists and ecologists, however, have focused a significant amount of research on these types of accidents, in terms of road kills [1,19,20,21,22,23]. Gaps in the literature on AVCs have reflected an inconsistent approach by both relevant authorities and specialists addressing the subject matter. If road mortalities are not properly investigated, mitigation measures are less likely to effectively address both driving safety and conservation concerns.

The increased demand and subsequent investment for road infrastructure have amassed globally [24,25]. Smith et al. [24] report that roads are predicted to be the largest investment area in transport infrastructure for, at least, developing countries, in the 2014–2025 period. Rising wealth in a country enables more of its population to buy and own cars. That being the case, many countries invest in large, extensive road networks [24]. If road infrastructure is expected to develop and expand, more space will be needed from surrounding natural areas. Animals occupying and maneuvering through these altered landscapes will inevitably enter the road network, raising the risk of an AVC.

The quality of road infrastructure is not necessarily proportional to AVC probability. Examples can be found in the United States of America (USA), where many AVCs and road kills are recorded over a myriad of roads, each with different infrastructure quality [26,27]. The USA has one of the most well-developed and expansive road infrastructures in some of its states, such as Florida, Colorado, Utah and Minnesota [28]. Due to the large number of vehicles occupying the road at a given time, several parts of the USA are also known for high road kill incidences. Florida ranks as one of the best states in terms of road infrastructure quality [29], and is regarded as notorious for its annually large numbers of reptile and amphibian road kills [30,31]. However, large road networks in poor condition are equally as dangerous for animals. California has one of the worst road infrastructures in the USA; approximately 45% of its roads are in poor condition [29,32]. In 2016, the state alone recorded 6737 road kills alone [15]. Poor management and maintenance of roads exacerbate the challenges animals and drivers face. If roads are not maintained up to expected standards, drivers will struggle to execute the necessary maneuvers to avoid an accident of any kind.

The recording of road kills is typically sparse of country-wide datasets. Specific roadways where AVCs commonly occur are often used as data collection points. Taiwan is one of the few countries to record road kills on a national scale. The Global Biodiversity Information Facility (GIBF) contains a database with records of a total 46,416 road kills from 2011 to 2017 for Taiwan, with an average of 6630.86 per year, in this period [33]. These statistics provide an essential source of analysis for AVC patterns on a national scale. Even so, local factors affecting the frequency of AVCs will vary across and within different roads in a country. Therefore, a large body of literature on AVCs and road kills typically focuses on a specific roadway within a locale, instead of all the roads in a country. Arévalo et al. [34] studied the spatiotemporal variations of amphibian road kills in a 4 km road segment in Costa Rica, namely, Costanera Sur. Their findings revealed that vehicle speed and traffic volume were important factors for road kills AVCs on that segment. In Nigeria, Halidu [35] surveyed road kills in two major routes of the Kanji Lake National Park; New Bussa-Lumma and Ibbi. For this study, weather, vehicle velocity, and visibility conditions were found to have the most substantial influence over AVC probability. The influential variables in the studies mentioned above can be used to advise mitigation for road mortalities in those areas and be compared to results from studies within and outside of their country. Identifying the influencing local factors for road kills over a particular stretch of road is, thus, significant for addressing the consequences of AVCs.

AVCs present many social, economic, and environmental consequences [1,23,36,37]. Road accidents involving animals can result in severe economic costs due to the resultant fatalities, injuries and property damages [1,2,15,28,38]. Vehicle damage and insurance claims are especially a challenge for the persons involved in the AVC [1,2]. The social consequences of these accidents are typically not evaded either. The aftermath of an AVC can threaten a society’s stability and functioning due to the compromised road safety and imposed costs. On the contrary, road kills not collected by people can serve as an important food source for scavengers, potentially improving ecosystem functioning [39]. Clark [40] explains that when scavengers consume road kills, they not only sustain themselves but reduce the potential number of neglected road kills disposed of in landfills. Despite these opportunities provided by road kills, AVCs themselves have continued to pose a serious and negative impact on the natural environment. Species population numbers are reduced, potentially compromising ecosystem functioning in natural areas, and subsequently, conservation efforts [1,10,21,23,41].

This paper reviews relevant literature regarding the nature and impacts of animal-related accidents (ARAs), as well as the consequent road kills. The paper aims to provide a cogent evaluation of existing knowledge on the subject matter. Subsequently, recommendations are made to address gaps in the literature, in order to advance future studies. The paper reviews six key focus areas; (1) road kill observations, reporting and recording (2) the significance of the spatiotemporal patterns of road kills locally, (3) the economic, social and environmental consequences of vehicle collisions with animals, (4) the current strategies used to mitigate animal-vehicle collisions and the resultant road kills, (5) the challenges that hinder the effectiveness of the current mitigation measures used to address animal-vehicle collisions; and (6) the use of geographical information systems and other technologies as a tool to uncover the magnitude and underlying patterns of accidents in involving animals, in order determine the most effective means to mitigate these events.

## 2. Review Methodology

The review paper identified and evaluated relevant literate about animal-vehicle collisions that resulted in road kills. Articles that focus on one or more of the primary topic areas were critically searched through all available research platforms. The primary databases used to obtain all relevant knowledge and information on these focus areas included Google Scholar (https://scholar.google.com (accessed on 9 January 2021)), EBSCOhost (https://www.ebsco.com/products/ebscohost-research-platform (accessed on 9 January 2021)), Scopus (https://www.scopus.com/ (accessed on 9 January 2021)), and ScienceDirect (sciencedirect.com (accessed on 9 January 2021)). In the database, search criteria included usage of the following keywords; “animal-related accidents (ARAs)”, “animal-vehicle collisions (AVCs)”, “clusters”, “carcass permanency”, “consequences”, “conservation” “economic impacts”, “environmental impacts”, “environmental management” “hotspots”, “hotspot analysis”, “influencing factors”, “kernel density estimation”, “location”, “modelling”, “prediction”, “road kill”, “road kill apps”, “road kill mitigation”, “road infrastructure”, “road mortality” “social impacts”, “spatiotemporal patterns”, “waste management” and “wildlife-vehicle collisions”. Journals focusing on content relevant to one or more of the topic areas were selected for review. Journals included, but not limited to, were the African Journal of Ecology (https://onlinelibrary.wiley.com/journal/13652028 (accessed on 9 January 2021)), the Journal of Applied Ecology (https://besjournals.onlinelibrary.wiley.com/journal/13652664 (accessed on 9 January 2021)), the Journal of Ecology (https://besjournals.onlinelibrary.wiley.com/journal/13652745 (accessed on 9 January 2021)), the Journal of Environmental Management (https://www.journals.elsevier.com/journal-of-environmental-management (accessed on 9 January 2021)), the Journal of Environmental Engineering and Landscape Management (https://www.tandfonline.com/toc/teel20/current (accessed on 9 January 2021)), the Journal for Nature Conservation (https://www.journals.elsevier.com/journal-for-nature-conservation (accessed on 9 January 2021)), the Journal of Transport Geography (https://www.journals.elsevier.com/journal-of-transport-geography (accessed on 9 January 2021)), the Journal of Wildlife Management (https://wildlife.onlinelibrary.wiley.com/journal/19372817 (accessed on 9 January 2021)), the European Journal of Wildlife Research (https://www.springer.com/journal/10344 (accessed on 9 January 2021)), and the Open Journal of Ecology (https://www.scirp.org/journal/oje/ (accessed on 9 January 2021)).

All articles were thoroughly assessed and selected for the review paper, based on their relevance to one or more of the aforementioned six focus areas. Although certain articles were included in the review section that they were most related and relevant to a significant body of literature overlapped with more than one of the key topic areas. That is, several articles were found to have integrated more than one of the key topic areas. Consequently, most articles have been included in more than one review section. The reference list of all articles has also been examined to find potential, additional literature to include in the review paper. Any article that was not related to least one of the topic areas was excluded from the paper. In total 144 sources were used in this review, including journal articles, peer-reviewed literature, books, theses, reports and conference papers. Additionally, websites and newspapers were included in the present paper.

## 3. Results and Discussion

### 3.1. Road Kill Observations, Reporting and Recording

Road kills can be recorded using specific observation techniques [42], such as identifying carcasses on foot, by transportation [1], or engaging the public to report road kills [4,43]. For example, in Tanzania’s Kwakuchinja Wildlife Corridor, Njovu et al. [44] completed 364 daily road kill surveys on foot, from 17 August 2014 to 16 August 2015, and covering a distance of 3094 km. This study recorded 82 road kills during the period mentioned above. Similarly, Coelho et al. [45] observed road kills on foot once a month for sixteen months in Brazil’s southern Atlantic Forest. They identified that 1433 road kills took place in this area. In addition, Husby [20] monitored road kills in Norway by driving a car along a 25 km stretch of road for a total of 617 days over five years. 121 road kills were recorded during this time. These findings suggest that observing road kills in the field can be time-consuming or spatially-restrictive for researchers and traffic management personnel alike.

Since the surveying of present road kills is potentially time-consuming and limits the spatiotemporal coverage of observations [43], many studies have used historical records; normally owned by transport and police departments, and conservation agencies [1,37,46]. Transport and police department personnel often record road kills when an AVC has been reported and caused significant damage to the vehicle or injury to the person/s inside it [1]. Some conservation agencies record road kills to determine the magnitude and composition of mortalities for certain fauna and their potential impacts [4]. These datasets can represent a large spatiotemporal range of the road kills that have occurred [47]. Bartonička et al. [1] obtained data for AVCs that occurred from October 2006 to December 2011 from the Police of the Czech Republic, for the aforementioned country. This data accounts for all AVCs that occurred in the country for more than five years, qualifying as a suitable source to analyse the rate of road kills. These records, thus, significantly save time and enable a more extensive, in-depth analysis of the spatiotemporal patterns of road kills. In support, Colino-RabanalandPeris [47] praise the use of historical records for its time efficiency and large spatiotemporal range. However, utilising such data does not usually provide user/s with recent recordings for the road being studied. To address this potential shortcoming; citizen science was introduced to incorporate more recent records while allowing for the same time-efficient and spatially extensive data acquisition.

Citizen science is the voluntary collection and analysis of scientific data by the general public [4]. A citizen scientist is someone who volunteers to collect or process data as part of scientific enquiry [43,48]. Members of the public are able to report the road kills they observe, to the relevant authorities or other observation systems, contributing to data for a given location or set of locations [49]. The more members of the public who participate in citizen science, the greater number of road kills likely to be reported. Above all, using citizen science increases the spatial extent of road kill reporting by enabling public members to report road kills from any distance and on any road [4,50,51,52]. The Taiwan Roadkill Observation Network (TRON) (https://roadkill.tw/en (accessed on 9 January 2021)) is a citizen science project run by the Taiwan Endemic Species Research Institute, allowing members of the general public to report road kills anywhere in the country [33,51,52]. When a person identifies a road kill, they can record the time and Global Positioning System (GPS) location of the animal, take a photo and then submit all this information online to the website [53,54]. The project staff then verifies this information before it is officially published on the network’s website as open source data [54]. Thedata’s open access reinforces the integrated efforts, employing engagement, between the general public, scholars, specialists, and relevant authorities. This demonstrates not only the vast spatiotemporal range of road kill data, but its utility in informing mitigation measures as well. Yue et al. [54] utilisedroad kill data from the TRON, focusing on snake mortality records from 2006 to 2017, throughout Taiwan. The records revealed that snake road kills were very high on roads in the low to mid-elevation forests. On that account, Yue et al. [54] advised that mitigation measures such as fences and passages should be prioritised primarily towards roads in low-mid elevation forests. 

The methods for recording road kills vary by technique and scale. Once road kill data is obtained, however, the aim should be to analyse the existing patterns and trends to determine the underlying, influential factors.

### 3.2. The Significance of the Spatiotemporal Patterns of Road Kills

This critical discussion informs how road mortality patterns reveal important trends and findings in the data collected during a given study. These patterns help determine how meaningful the clusters of road kills are [55]. Road kill clusters explicate on which particular portions of the road, and during what periods of the day, animals are more frequently colliding with vehicles. In other words, these clusters are incidences of different road kills located spatially and/or temporally close together. The spatial proximity of AVCs demonstrates which road kills are located close together, over a specified locale. The spatial component of road kill clusters, thus, explicates where road kills occur, and by extension, where they occur most frequently [1,21]. Temporal proximity, on the other hand, expresses closeness of road kill occurrences to one another over time [37]. The temporal component of road kill clusters reveals significant patterns and subsequently informs when animals are most susceptible to being hit by an oncoming vehicle [21]. Meaningful spatial clusters are referred to as hot spots, whereas their temporal equivalents are hot moments; which jointly reflect the spatiotemporal patterns [37,55]. These clusters can be identified and analysed from any spatial scale. However, Bartonička et al. [1] asserts that road kill clustering is explained specifically by local factors, such as traffic volume, vehicle velocity, and distance to specific land cover. There has, consequently, been a specific focus on the local influences over road kill clusters on transport routes, in the majority of literature. Spatiotemporal patterns are a popular topic in geographic information systems (GIS); including ArcGIS. The analysis of road kill clustering and spatiotemporal patterns, using a GIS application, is discussed in more detail in Section 3.4.5.

In particular, spatiotemporal patterns of road kills enable researchers and relevant authorities to uncover the influencing factors. The reason whycertain road kills are spatially and temporally located close to one another on specific portions of a roadway may be explained by a number of local factors. These factors will differ from one locale to another, and some are more prominent in previous studies than others. The road, itself, is a factor with both direct and indirect influences on AVC incidences [14,56,57]. The influences analysed in this section are, proximity; road infrastructure; traffic volume and vehicle velocity; driver awareness; landscape; climate and weather conditions; and animal behaviour.

#### 3.2.1. Underlying Factors in Spatiotemporal Patterns

Spatiotemporal patterns reveal a plethora of factors influencing the existing numbers and clusters of road kills. These factors are often categorised into specific groups based on a specific system of characterisation used by the researcher/s. Different studies have grouped factors according to their own systems and criteria. For instance, Kazemi et al. [2] identified four main elements of road kills; that is, the major groups of factors for road mortalities. These are:Animal-related, behavioral and biological factors: Species population density, animal behaviour, foraging nature, mating, migrating, breeding and rutting.Habitat-related factors: Habitat patches distribution, resources availability.Weather conditions: Temperature, precipitation, cloud cover, wind.Anthropogenic causes: land use, traffic density, driver-related conditions.
On the other hand, Bartonička et al. [1] groups the factors potentially responsible for AVCs into three categories:
Species ecology and behaviour: Sex, age, dispersal, habitat use, mating, breeding and rutting.Traffic factors: Vehicle velocity, visibility and traffic density.Environmental factors: Presence of natural corridors, fragmentation.

Numerous other studies have followed suit [12,22,45,58], using their own classification systems. Regardless of how the road mortality factors are grouped, typically, no single influence operates in isolation. Several studies have each determined a number of relationships between different factors.Therefore, two or more factors can integrally influence the spatiotemporal patterns of road kills. The study by Kazemi et al. [2] identified that weather conditions influence visibility and animal behaviour, on the road. Poor weather conditions often mar the ability of drivers or animals to see one another on the road early enough to avoid a collision. Although road infrastructure is a factor for mortalities [46], it serves as a significant context for the interrelationships between other factors. For instance, roads alter the natural landscape and cause habitat fragmentation. Animals inhabiting these fragmented natural areas typically do not have access to resources necessary for their survival; in the immediate patch they occupy [12]. As a result, those species, which require a particular food or water resource in a patch segregated by these networks typically have to maneuver across these roads to fulfill a particular biological need [59,60]. The proximity of these animals from the patch they currently occupy, to the one they seek travelling to, typically influences the likelihood that they will encounter a vehicle [1,23]. As a result, these animals are susceptible to a vehicle collision. On the other hand, traffic density and driving speed will influence the extent of this risk [2,23,61]. As a result, when and where road kills occur is the product of several integrated influencing factors. For that reason, a myriad of interactions between different factors may be considered when analysing the spatiotemporal patterns of road kills for a given area. The spatiotemporal patterns revealed in AVC datasets are, consequently, a necessary indicator of where to start considering certain factors. These factors particularly help determine which questions to ask about the road kill occurrences.

##### Spatial Proximity

The nearness of road kills to one another, of course, explicates the spatial patterns of AVCs for a given stretch of road. Notwithstanding this fact, proximity equally expresses a significant role in influencing the magnitude and spatiotemporal patterns of road mortalities [62]. In fact, spatial proximity interacts with a myriad of factors that influence road kill rates and patterns [62]. Previous studies have found that the proximity between animals, the road network and influential AVC variables is a significant factor [1,37,45,62]. The distances at which animals are located from particular environmental variables, such as land cover, often affect their movement, and thus, risk being hit by an oncoming vehicle [45]. From 2006 to 2011, Bartonička et al. [1] found that most road kills occurred at distances of less than 350 m to forests in the Czech Republic. Accordingly, habitat and resource requirements locates some animals closer to roads and subsequently places them under increasing risk of a vehicle collision [62]. Road kill hotspots are, as a consequence, often located within close proximity to desirable habitats for the species composing these mortalities [63]. A good case in point is that, for animals such as reptiles and amphibians, nearness to a perennial waterbody is normally related to significantly high clusters of road kills [5,45]. Animals’ niches located closer to the forest interior have a significantly lower probability of colliding with a vehicle. On the other hand, for some species, close proximity to land cover, such as urban and residential areas is associated with fewer incidences of road kills [64]. Proximity-based influences over road kill presence are, nonetheless, not limited merely to the subject of land cover. Kim et al. [37] studied the relationship between the number of road kills and their proximity to ramps. The authors observed that on highways, fewer road kills tend to occur closer to ramps where traffic conditions are relatively stable. The road network, traffic volume and vehicle velocity play an equivalently significant role in probability of AVCs, in terms of their proximity to ramps or other features of the road [37].

##### Road Infrastructure

There are several types of road infrastructure; each classified using a different system [65]. Road classification systems often differ between countries or regions. Road class alone cannot directly influence the number of road kill occurrences unless correlated with another factor such as traffic intensity [1]. The type of road reflects its structure, function and location. The structure includes its materials, signs and markings. A considerable amount of research has been pursued on comparing AVCs in paved and unpaved roads [45,66,67]. The materials, themselves, do not typically affect the number of road kills in an area. However, the design of the road can determine the habitat characteristics for which species will use it [14]. That being the case, some species are more susceptible than others to colliding with vehicles on certain roads. Brock and Kelt [68] compared the abundance of kangaroo rats (*Dipodomys stephensi*) on two different roads. The study concluded that kangaroo rats used dirt roads more often than gravel ones. This is owing to *Dipodomys stephensi* preferring roads with sandy substrates and finding those with gravel material to be uncomfortable to walk on [68]. Accordingly, the more frequently kangaroo rats use dirt roads, the more likely they will be hit by an oncoming vehicle, at least compared to the gravel type. Upon investigation, Bitušík et al. [69] also found that carnivorous animals were the most likely to collide with vehicles on the raised segments of two roads in the southwestern part of the Banská Bystrica Province in Central Slovakia. In addition, other factors, such as traffic intensity, presence of barriers, and vehicle velocity may influence the extent of AVC risk for a particular road design [1]. Road signs and markings, on the other hand, can influence the chances of an AVC [70]. Signs and markings that are absent, unclear, or simply have not communicated the necessary action for the drivers to execute, increases the risk of a collision with an animal [3].

The size of the road infrastructure can be both explicative and causative of the number of animals colliding with vehicles [71]. Often, certain sections of a stretch of road have more road kill occurrences than others; hence the presence of clusters [1,69]. Bartonička et al. [1] observed that road width was one of the most influential factors contributing to road kills in 1250 km of motorways in the Czech Republic. Particularly, roads with a width greater than or equal to 7 m influenced the likelihood of AVC clusters. Similar to road design, however, the integrated influence of other factors can redirect the effect of width on road kill risks. A case in point; in the study by Bitušík et al. [69], the authors concluded that small rodents were killed by oncoming vehicles more often on narrow roads, where canopy was located nearby and above road segments. Although, these narrow roads experienced low traffic intensity, the aforementioned road characteristics encouraged greater movement among small rodents. Road infrastructure can, thus, influence the surrounding environment. As previously mentioned, though, how traffic volume and velocity interact on the road infrastructure has an equal, if not more significant role influencing incidences of AVCs.

##### Traffic Volume and Velocity

The number of vehicles crossing a road, or segment of it, exerts some influence on the number of road kills over a specified period of time [72,73]. Traffic intensity varies with time, and on different types of roads [1]. Holidays periods can often result in increased traffic volume [74]. Subsequently, high traffic volume within this period results in a higher number and rate of road kills [12]. The location or function of a road can often infer the extent of traffic volume. According to Bartonička et al. [1] and Kuhn [75], intermediate roads experience higher traffic intensity than major highways or local access roads. Interestingly, roads that experience a high traffic volume do not always result in high road kill counts [57,62]. In southwestern England, a study identified that badgers (*Melesmeles*) were often discouraged from using major roads with the highest traffic intensity [73]. Furthermore, this road was found to experience the highest vehicle speeds. As a result, high traffic volume and vehicle speeds conversely dissuade animals from utilising the road. In that case, such variables can act as a barrier to animal movement and reduce road kill risk.

Notwithstanding the aforementioned findings for traffic volume and vehicle speed, Rosell et al. [76] attribute the increased velocity of cars as one of the main causes of the high number of AVCs in European countries. Similarly, in India, Tayade et al. [3] noted that one of the influencing factors of Indian palm squirrel (*Funambulus palmarum*) road mortalities was high-speed driving. As with traffic volume, vehicle velocity is often influenced by the time of day and the specific road on which the cars drive. Regarding the former, driving speeds during the day and night may vary and, as a consequence, differentially impact the number of road kills that occur between these two distinct periods [70,73]. The latter is typically dependent on the speed limits assigned to the road or particular portions of it [3]. The general assumption is that speed limits positively correlate with road kill rate [77]. Accordingly, the higher the speed limit for vehicles, the larger the number of animals that are likely to be involved in vehicle collisions [78]. However, some studies have shown contrary findings. Husby [20] observed that as speed limits increased on a stretch of road in Norway, the road kill rate and the probability of certain bird species being hit by an oncoming vehicle decreased. In the context of the birds sitting on the road, as vehicle velocity increases, the likelihood of the bird flying away increases [79]. The type of road alignment has previously been found to influence the extent to which increased vehicle velocity impacts the risk of AVCs [20]. Hernandez et al. [80] identified that on curved roads, as vehicle speed increases, the number of bird road kills rose. Road alignment interestingly, shares an important relationship with driver awareness [20,58,80].

##### Driver Awareness

The vehicle’s driver can exert considerable influence over the probability of an AVC occurring [74]. Of course, if an animal simply springs onto the road and in the immediate face of oncoming vehicles, preventing an AVC becomes far more improbable. Conversely, if a motorist is able to identify the animal well in advance and execute the necessary action, an AVC becomes far less probable. Adherence to speed limits and rules of the road can substantially reduce the number of road kills [3]. Therefore, good driver awareness is necessary for reducing the number of road kills. Several other factors, however, can compromise the ability of drivers to prevent an AVC. Visibility conditions need to be optimal for the motorist to avoid hitting an animal with their vehicle [1,2]. For that reason, the factors that influence driving visibility conditions will subsequently affect the risk of AVCs. The road’s alignment can strongly influence the extent to which drivers can see the animal well in advance before a collision occurs [58]. For example, hills or curves on a road segment can reduce driver visibility and increase the risk for an immediate collision [81]. If roads are poorly lit, or there are no proper warning signs that inform drivers to reduce their speed and remain vigilant on AVC-prone roads, then more accidents will arise [3,82]. Particularly during the evening, poorly lit settings can exacerbate the probability of AVCs [82]. Poor weather conditions can further negate the effectiveness of road signage and lighting [2]. Heavy rainfall, mist, and fog reduce driver awareness and increase the number of road kills. Driver awareness can, as a result, vary during different times of the year, based on the prevailing conditions.

##### Landscape

Road kills are both a product and a symptom of the altered landscape [4,14,57,83]. Existing landscape characteristics influence the spatiotemporal patterns of road kills, due to the specific habitats and resources animals require [62]. Landscape characteristics are altered by changing land uses. Roads, themselves, are a land use [84]. Moreover, road infrastructure development is one of the influencing factors for other land use changes, over time, in a given area [85]. The location and function of a road can, thus, infer its surrounding land uses. Emerging land uses such as agricultural activities, residential areas, and roads have a significant impact on animals’ natural habitats [84]. As mentioned in the introduction, the development of roads in the natural landscape causes habitat fragmentation. Similarly, the surrounding land uses further fragment these areas [86,87]. Many animals maneuver across the fragmented patches. When more animals cross the roads between these patches, there is a subsequently greater risk of AVCs. Findings from Kazemi et al. [2] show that the development and renovation of the Asiaie Highway had resulted in a high number of animal road mortalities in Iran’s Golestan National Park. As a result, land uses alter the natural landscape and can subsequently influence the number of animals involved in AVCs.

Habitat fragmentation, thus, problematises the ability of animals to reach their resources, by separating both entities in disconnected natural patches [9]. The closeness of roads to habitat patches suggests that animals occupying and maneuvering between these areas tend to be killed more by oncoming vehicles. Natural corridors are the areas of habitat that connect animal populations that have been separated by roads and other land uses [88]. Natural corridors include forest edges and streams [1]. Forest edges have been particularly significant in conservation efforts that address habitat fragmentation [88]. It is important to note though, that the suitability of these edges varies among different animals; as the environmental conditions differ from those in the forest [89,90]. Bartonička et al. [1] stated that the length of the forest edge increases the presence of AVC clusters. Animals that use these forest edges to move between habitat patches place themselves under an increased risk of being hit by an oncoming vehicle.

##### Climate and Weather Conditions

Unfavorable climate and weather conditions have a profound impact on road kill presence [2]. Poor visibility conditions owing to heavy precipitation, mist or fog can distort the visual perception of both drivers and some animals, thereby, increasing the probability of an AVC. Heavy rainfall during the day or night can also discourage the movement of some animals; thereby reducing AVCs [91]. In contrast, favorable climates can encourage high levels of species abundance and richness in habitat patches [92]. As a consequence, a higher number of road kills can occur when these species are active near or on the road. Notably, different species will have differing responses to climate and weather conditions. A study by Carvalho et al. [93] in the BR-050 highway of Southeastern Brazil, determined that reptile road kills increased during those periods of heavy rainfall, as opposed to drier spans.

Seasonality is typically associated with the temporal patterns of road kills [1,45]. Depending on the locale, the influence exerted by different seasons has varying effects on the frequencies and rates of road kills. In da Rosa and Bager [9], seasonal bird road kill patterns in southern Brazil were concentrated for summer and autumn; 265 and 202 mortalities respectively, from the total 671. Kazemi et al. [2] identified the spring and summer months as having significantly high road kill counts in the Golestan National Park. Particularly in summer, these species are found in large numbers in the seasonal wetlands [94]. Landscape characteristics, niches and animal behaviour are, hence, integrated with seasonality to influence the number of road kills. Temperature, rainfall and photoperiod have been found to be associated with seasonality in influencing road kill frequency and rate [45,95]. Another study, which occurred between June 2002 and October 2003, and took place near the Iva State Park in Brazil, found that anuran road kills were high during the summer; temperature being the most important, associated variable [45]. Favorable, warmer temperatures were conducive to anuran activity and subsequently increased road mortalities [45,96]. Precipitation can significantly affect the activities, movements, and locomotive performances of amphibians, particularly [96]. When rainfall increases, anuran activity rises and, as a consequence, more of these amphibians are hit by vehicles on the road [45,96]. Temperature and precipitation, therefore, have a direct, significant impact over the number of amphibian road kills. During summer, the average number of hours of sunlight is comparatively higher and dramatically impacts the number of road kills [95]. Interactions between photoperiod and temperature result in higher reproductive rates [97]. The photoperiod expresses to anurans the season for which to mate and breed [95]. The temperature interacts with this photoperiod, to trigger hormonal stimuli that promote breeding [98]. Seasonal variations in road kill patterns are typically influenced by the predominant animal activities during these periods. Migratory, mating, breeding, rutting, and hunting seasons all demonstrate significantly high road kill mortalities [23,37,58,74]. For example, male leopard cats expand their home ranges during mating season and accordingly experience a higher probability of being involved in a vehicle accident [37]. Romin and Bissonette [58] note that mule deer road kills are most frequent during breeding and hunting seasons at the Jordanelle Reservoir in the US. Therefore, animal behavior during a particular season can impact the frequency and rate of AVCs.

##### Animal Behaviour

The behaviour of animals has a strong influence over their probability of being hit by a vehicle [4]. Where animals migrate, forage, territorialise, mate, breed, or disperse, express their behavior, which in turn influences their movements [1]. A good case in point is where a herd of animals, such as ungulates, is separated by a road. The animals at the back of the herd will typically follow the dominant members who have crossed the road; thereby increasing their risk of being involved in an AVC [74]. Animal behaviour, evidently, influences the specific movement patterns of different species [1]. Valerio et al. [23] differentiate an animal’s movement-based behaviours into two categories; daily and dispersal. Daily movements include those for the animals’ foraging and mating activities, and dispersal is when animals colonise new territories [99]. The active, foraging behaviors of some animals are often dictated by the time of day [73]. The nocturnal behaviour of badgers, wild boar and certain amphibians has often resulted in the road mortalities of these species being concentrated during the evenings [1,9,45,73]. The increase of road mortalities at night is equally influenced by other factors such as traffic volume [23]. In the evenings, if traffic increases beyond a certain volume, some nocturnal animals are more likely to avoid the roads [73]. A study by Jacobson et al. [100] determined that certain animals’ specific behavioral responses to oncoming vehicles could influence the likelihood of an AVC; thus, some species respond more effectively to these dangers than others. These responses, in consequence, produce the spatiotemporal patterns of the road kills. Williams et al. [67] confirmed that during dry seasons, from the beginning of April till the end of September, serval (*Leptailurus serval*) road kills are significantly higher than during the rest of the year. A total of 54 out of the total 86 recorded road kills recorded were during the dry season. When water is scarce, servals and other carnivores expand their home ranges and often cross roads through their expanded territory [101]. Mating season can affect animal behaviour and habitat use [37].

#### 3.2.2. The Socioeconomic Impacts of Animal-Vehicle Collisions

The financial consequences of AVCs can be cumbersome for owners of the vehicles involved, who must bear the costs for vehicle damages [38]. On highways and selected main roads in California, wildlife-vehicle accident costs amounted to approximately 276 million US$ in 2016 [15]. More than US $94 million of this amount comprised property damages, suggesting strong inflictions to the financial security of vehicle owners. The situation is equally problematic when injuries or fatalities are incurred. The São Paulo State in Brazil averages R $67,048 in costs per AVC when injuries or fatalities have occurred [10]. Furthermore, in Brazil, road administrators are held liable for the vast majority of AVCs, compensating R $2,463,380 per year for victims in the São Paulo State alone [10]. In 2003, ARAs resulted in over 200 million € in France alone [26].

AVCs that result in severe vehicle damages, serious human injury or fatality, can subsequently affect social lives [2]. From 2007 to 2016, 152 patients were admitted to Victorian hospitals in Australia due to animal-related vehicle crashes [16]. For the São Paulo state in Brazil, 18.5%, or approximately 483 of an average of 2611 AVCs per year resulted in human injuries or fatalities [10]. Individuals who have been seriously injured in ARAs may not be able to participate in certain activities. Those individuals whom have been killed in ARAs may render their households to a more vulnerable state where the remaining family members struggle to maintain their social livelihoods.

#### 3.2.3. The Environmental Impacts of Animal-Vehicle Collisions

AVCs can be disastrous for some animal populations. The behavioural-physiological characteristics of a species influence not only their probability of being hit by an oncoming vehicle but the severity of pressure these events place on their population numbers [12,42,102,103]. ARAs of vertebrate species lacking avoidance behaviour, having greater mobility, a larger home range and body size, lower reproductive rates and population densities, are of relevant concern to conservationists and other relevant authorities [104]. Several mammalian species are, as a result, part of national biodiversity concerns for different countries [83]. However, amphibians and reptiles are some of the world’s most endangered species [42]. The behavioural traits of these vertebrate species render them increasingly susceptible to AVCs [83]. The resultant loss of these species numbers from road mortalities, amongst others, stands to threaten their populations. Howbeit an animal’s vulnerability or endangered status, their reduced numbers stand to threaten the functioning of the ecosystems in which they function [3]. Even the reduced population size of common species can be problematic for ecosystem functioning and services [103]. Indeed, Gaston andFuller [105] assert that the proportional reduction of common species can cause large losses of individuals and biomass, which compromise the ecosystem’s stability and functioning. Road mortalities can even act as a barrier to animal movements, suppressing the full extent of dispersal rates and other evolutionary processes [100]. A lesser-discussed environmental impact of AVCs is that the resultant road kills, themselves, notably, contribute to air pollution levels. When relevant authorities collect road kills, they are usually thrown away into landfills [40]. These decomposing carcasses release methane emissions in landfills, which are often already filled with other organic by-products [106].

### 3.3. The Current Mitigation Strategies Available to Address Animal-Vehicle Collisions

The growing attention of road kills in academic discourse has prompted more robust engagements among the relevant parties responsible for managing road safety or conservation. Implementing sufficient mitigation strategies for preventing AVCs potentially eliminates its impacts. This section analyses the mitigation measures currently used by relevant authorities, and includes managing road conditions, the use of overpasses and underpasses, and fencing.

#### 3.3.1. The Managing of Road Conditions

In several AVC studies, it has been observed that increased vehicle velocity leads to significantly high road kill counts [1,3,76,77]. High vehicle speeds are especially problematic for animals in the evenings [45,73,82]. For this reason, the determination of nighttime driving speeds and distances have been targeted to identify the best decision-making for the driver to avoid hitting an animal on the road [107]. When animals are detected from a specific distance, the drivers need to follow the cue to control their speed, in order to prevent the AVC. Structures added to the road infrastructure to prevent AVCs include wildlife crossings and fencing.

#### 3.3.2. Wildlife Crossings

Wildlife crossings are structures built to safely and strategically guide animals from one area to another [108,109]. There are three main groups of wildlife crossings; overpasses, underpasses, and road-level crossings [108,110]. Overpasses and underpasses allow animals to cross over to another habitat, without any risk of being hit by an oncoming vehicle. This is accomplished by building these structures above and below the highways that would be utilised by animals. That is, overpasses are built above the road, and underpasses are built below it [110].

Structurally, overpasses can be crossings over a tunnel, landscape bridges, or large green bridges [110]. Crossings over a tunnel are, as implied, structures built over tunnels for usage by animals. Landscape bridges exceed 100 m in width and should typically be covered with natural vegetation [110]. Green bridges are big overpasses, characterised by natural ground and vegetation covering, as seen in (Figure 1, [111]). The design of these structures primarily depends on the magnitude and composition of species they are trying to protect [108]. The suitability of wildlife crossing structures, then, usually depends on the species of animal [64]. Cramer [112] monitored two overpasses in Utah specifically designed for big game, such as mule deer. She noticed that bridged wildlife crossings (including overpasses) had a higher mitigation success rate than culverts for mule deer. That is, bridges had an average 87% success rate for mule deer, compared to the 74% success rate using culverts. Moreover, shorter wildlife crossing culvert lengths were associated with reduced numbers of AVCs. Wildlife crossings may be designed to target endangered species such as desert tortoises or salamanders [108]. Overpasses and underpasses improve habitat connectivity by linking natural patches [82]. This eliminates the need for animals to use a road as a crossing mechanism. In consequence, fewer AVCs result as more animals use the overpasses and underpasses, as opposed to the road. However, the effectiveness of wildlife crossings will depend on where they are built [23].

Underpasses are underground passages that include tunnels and culverts that have a width between 0.5 m and 2.0 m; see (Figure 2, [111]). Underpasses are usually aimed at small, nocturnal animals such as badgers and hedgehogs [113]. These structures may require a guiding system such as a wire net or fencing to lead the animal into the passage.

Crossings on the road surface are the most common type of the three categories. Improved road signage and signals about the possibility of animals appearing on the road serve as one of the chief methods of reducing AVCs [110]. These include speed limits and information signs; see (Figure 3, [114]).

#### 3.3.3. Fencing

Fences act as important barriers that prevent animals from getting onto the road. That is, fencing contributes significantly to road avoidance by animals [115]. Wildlife exclusion fencing is considered the most effective means to reduce AVCs [116]. A study by Rytwinski et al. [117] confirmed that fencing reduces wildlife-vehicle collisions (WVCs) by 54%, regardless of whether they are combined with wildlife crossings. This finding is significant because it reflects the cost-effectiveness of fencing as a road kill mitigation measure.

### 3.4. Challenges That Compromise the Effectiveness of Strategies to Address AVCs

The success of strategies to address AVCs can be impeded in two ways. Firstly, the recording and quality of road kill data can be compromised, hindering its utility in selecting or assessing mitigation measures [1]. Secondly, the mitigation measures may suffer from internal or external influences that compromise their effectiveness in preventing AVCs [116]. The main challenges associated with hindering the effectiveness of road kill prevention or reduction strategies, are discussed below.

#### 3.4.1. Inaccurate and Unreliable Reporting

To manage, reduce or prevent AVCs, road kill data needs to be accurate and sufficiently explicate the spatiotemporal patterns on a stretch of road [1,118]. Reporting of road kills is inevitably prone to some human error, such as missing a road kill during observations or recording [45,119]. These errors are normally not detrimental to the dataset’s utility, since they are usually infrequent [119]. Nevertheless, some major challenges compromise the reliability of road kill data for both developing and evaluating mitigation strategies.

AVCs are typically reported when they result in some form of vehicle damage or human injury [1]. AVCs that do not pose any harm to humans are, at least, relatively less reported. Vehicle damages or human injuries normally result if the vehicle hits a significantly large animal such as an ungulate [10]. Therefore, smaller animals such as amphibians, small reptiles and rodents are often at risk of being underreported [42]. This challenge is hampered by the fact that many of the road kills of smaller animals may not last long enough, on the road, to be discovered and reported [91,120]. A study by Santos et al. [119] identified that road kills of animals with a body mass of less than 100 g had a lower detection than those that were heavier. Husby [20] established that in Norway, road kills of small birds did not last on the roads, as long as larger birds.

Generally, any road kill is expected to last on the road for one to two days before they are removed by a scavenger or other external force [39,91]. Three factors influence the carcass permanency time of road kill; the presence of scavengers, weather and traffic volume [39,91]. Scavengers inevitably speed up road kill removal before it can be recorded [12,121]. Ratton et al. [39] observed that diurnal scavenger birds such as black vultures (*Coragypsatratus*), active near highways, significantly sped up the carcass removal rates on the roads. Additionally, the type of land cover near or surrounding the road can influence the number of scavengers, and, in turn, the removal rate of road kills. Santos et al. [119] discovered that areas with significantly high levels of savannah coverage and habitat had significantly low carcass persistence. This is attributed to the rich and diverse community of scavengers that inhabit savannah dominant habitats, as opposed to those lacking this land cover.

Weather has varying effects on road kill persistency, depending on the type of condition. Higher temperatures and dry conditions can desiccate road kills and increase their persistence on the road [119]. Yet, there are still humid conditions, which soften the carcass’s body, thereby increasing roadkill removal rates [91]. Heavy rainfall often deters scavengers from the roads, thereby perpetuating carcass persistence [61,91]. Besides, these heavy rains can also speed up the degradation of the road kill and wash away the remnants of its body [119].

Daily surveys conducted by Santos et al. [91] determined that increased traffic volume removed the road kills at an even faster rate through splatter and dismemberment of the carcasses. Moreover, the study concluded that smaller animals are removed at a faster rate than larger fauna. In contrast, some animals that have been hit by oncoming vehicles might not necessarily die on the road; in fact, they may suffer from a fatal injury after being hit by a vehicle on the road, and then travel elsewhere to be discovered dead [12]. Since road kills are normally reported on the roadway, the count of mortalities risk being inaccurate; so, if the animal has not suffered a fatal injury and the involved vehicle is unaffected, neither a road kill nor an accident is typically recorded. Hence, details of the accident’s occurrence may be neglected by the relevant authorities, and serve no potential utility in advising or assessing mitigation planning for the AVCs that result in road kills.

#### 3.4.2. Methodological Challenges

On a given stretch of road, if differing methodologies are each employed for a different session of observations and separated by long time intervals, they can compromise animal mortality results [1]. Surveys implemented at different frequencies and durations can miss significant road kill counts on the days or periods where they are not used [1,119]. As a result, this can obscure the temporal distributions in the data, due to missing road kill counts or clusters. On the testimony of Santos et al. [119], road kill hotspots and hot moments are negatively correlated with increasing time intervals between data collections, subsequently missing these clusters.

The taxonomic-based identification of AVC clusters is popularly used in many road kill surveys and observations [12,34,64,122]. Strict focuses on basing road kill counts or clusters to specific taxonomic groups overlooks the spatiotemporal patterns of mortalities [1]. This challenge is exacerbated by many smaller species’ carcasses not remaining on the road long enough to be observed and recorded [64,91]. If the spatiotemporal patterns of road kills are not accurately expressed in the data, optimal mitigation strategies cannot be determined. For that reason, a significant number of road kill studies compromise the importance of their results and what these findings are able to inform. Equally problematic is the challenge of insufficient road kill data, unable to serve as a suitable measure of the effectiveness of current mitigation procedures.

If road kill data is inaccurate, and, unreliable, mitigation measures are unlikely to be effective [1,123]. The procedures, themselves, are necessary to prevent as many AVCs as possible. Currently implemented mitigation measures still face some challenges due to prevailing external conditions within the context in which they are used [116].

#### 3.4.3. Poor Enforcement of Traffic Laws

Some road markings and signs are designed to dictate appropriate driver actions in order to prevent or reduce AVCs [1,3]. Tayade et al. [3] advocate the posting of speed limits, in order to address the challenges associated with AVCs. Speed limits do not typically reduce road kills if the animal is not easily identified by the driver [20]. The distance between the animal and oncoming vehicles plays an important role in determining probability of an AVC occurring [74,107]. Detection distances, specifically, are an important factor in determining the probability of an AVC. In Tasmania, the detection distances of animals differed with fur brightness; evidently species-specific [107]. Accordingly, driver awareness will play a significant role in preventing AVCs. Some driver attitudes also reflect a lack of awareness of animal road mortalities, which further emphasises the weak enforcement of traffic laws [3]. In such instances, the posting of road warning signs are unlikely to mitigate AVCs. In southern Michigan in the USA, 1635 drivers responded to a questionnaire by Marcoux andRiley [124], in regard to deer-vehicle collisions. Only 39% of the respondents said they would definitely reduce speed upon seeing a deer-crossing sign. Lee andCroft [125] have asserted that driver awareness is paramount to the success of AVC prevention. Kioko et al. [126] conducted a study in Northern Tanzania, interviewing drivers who frequently used the 40 km and 35 km transects on the Makuyuni-Babati, and Karatu-Makuyuni roads, respectively. Their findings reflected that drivers were unaware of the species most frequently hit by oncoming; that is, the majority of road kill victims were birds, but large mammals were far more recognised by motorists. Moreover, Kioko et al. [126] suggests that drivers intentionally hit those species which cause less damage to their vehicles, such as birds, dogs or frogs.

#### 3.4.4. Counterproductive Fencing

As stated by Clevenger et al. [115], AVCs can be clustered at fenced ends. When animals enter fenced rights-of-way from fence ends, they can become trapped between the road and this barrier, increasing their risk of being hit by an oncoming vehicle [116,127]. For instance, a 2 km road segment centered at the fenced ends of the Trans-Canada Highway, had a significantly high AVC rate [115]. Jensen [116] states that fencing alone worsens the fragmentation effect of road infrastructure. As a consequence, fences that are not combined with wildlife crossing structures exacerbate the challenges faced by animals; as they try to maneuver from one habitat patch, to another. The presence of such an anthropogenic barrier, in consequence, impedes on the movement of animals and is counterproductive for reducing road kill numbers. In Utah, USA, Cramer [112] determined that there were significantly higher mule deer success rates when wildlife crossings and fencing was combined as a mitigation measure, compared to when they were not. Additionally, on Route 175 in Quebec, Canada, road mortalities for medium-sized animals (such as porcupines) and red foxes were found to be higher in fenced ends than unfenced road segments [128].

#### 3.4.5. Geographic Information Systems (GIS) Used in Road Kill Studies

GIS has been a useful tool and application in analysing road kills in space and time [1,2,15,37,42,129]. Notwithstanding the significance of temporal patterns, spatial analysis is one of the most commonly used methods for assessing the road kill patterns [1,2,34,37,129]. Spatial analysis can be defined as a means of observing the geographical patterns of data, and analysing the relationships between events occurring in a given space [130]. Consequently, spatial analysis can be used to specify locations within the geographical space in which particular events occur and assess specific distribution patterns using map visualisation, as demonstrated by Nyoni [131]. There is a range of tools and techniques used in spatial analysis, which include Kernel Density Estimation (KDE) and hotspot analysis, amongst others. One of the underlying principles for conducting a spatial analysis on road kills is the concept of spatial autocorrelation. Spatial autocorrelation considers how well object or event correlate with other nearby objects or events across a given spatial location [132]. These objects or events are represented by values in spatial analysis [55]. If similar values are located near each other, there is positive autocorrelation; if very different results are located nearby, then negative autocorrelation occurs [132,133]. Although the significance of similar objects being near to one another is relevant understanding road kill patterns, the temporal component of these events offers equally important findings [1,74]. Several GIS tools and applications consider these temporal patterns of road mortalities [1,2,12,34]. Analysing the spatiotemporal patterns of road kills is, thus, enhanced with use of certain tools, techniques and analyses in GIS. Certain apps (such as “Road kill” and “RoadWatch”) that enable more effective reporting of road kills can be used in conjunction with GIS to better assess road mortality patterns, and subsequently improve selected mitigation strategies.

The KDE method has been used by Bartonička et al. [1] to identify where AVC clusters occur along roads in the Czech Republic. The study used the KDE+ approach to determine the presence or absence of AVCs in a cluster [134]. This builds on KDE [135], and estimates the probability density function of the underlying road kill data. The probability density function is the statistical expression that defines the likelihood of an outcome for a discrete random variable, as opposed to a continuous one [136]. A challenge is that KDE produces a range of local maxima that are not differentiated from one another, because of the absence of a defined threshold. The absence of an objectively defined threshold result in clusters that are not differentially significant. In response, Bartonička et al. [1] resolved this problem by introducing random Monte Carlo simulations (see [137]), selecting only significant clusters and ranking each. Consequently, meaningful clusters were identified and the factors influencing these patterns could be determined. Clusters that are most meaningful can, further, be used to identify which portions of the road are most prone to AVCs [1]. The study noted that 27.4% of AVCs occurred in clusters.

Hotspot analysis has been used in several AVC and road kill studies. WaetjenandShilling [15] mapped road kill hotspots across all roadways and highways in California, and determined the statistical significance of each of them. Wilson [129] conducted a hotspot analysis to identify critical areas of road kill occurrences in Southern California. In identifying the nearness and patterns of points (road kills), the aforementioned author was able to locate significant clusters of road kills. Eight hotspots were identified in the study, with seven of these showing high concentrations of road kill in urban land cover. This is an example of how hotspot analysis can identify important areas for mitigation. BissonetteandCramer [138] performed a hotspot analysis in a GIS to identify high concentrations of road kills. They subsequently provided safe wildlife passages on identified high-risk roads. Although hotspot analysis can inform state decision-making for mitigation measures, WaetjenandShilling [15] have noted the limits of its utility. These authors have previously observed that the identified hotspots only reflect reported AVCs and road kills, as opposed to the totality of that which occurred. Accordingly, the larger the number and scale of road kill observations, the more potentially reliable these records will be for better informing mitigation strategies.

#### 3.4.6. Using Smartphone and Web Apps for Road Kill Observations

Several web-based road kill reporting applications incorporate smartphone devices [139]. Participation of the public, for a larger scale of road kill reporting has been greatly encouraged through the use of apps [4,43]. In the UK, the Project Splatter app recorded 15 631 road kills in 2019 [140]. Two apps, both developed in India, by separate conservation non-government organizations (NGOs), are freely available for use by the general public for reporting road kills; “Road kill” by the Wildlife Conservation Trust, Mumbai and “RoadWatch” by the Wildlife Trust of India, Noida [4]. “Road kill” aims to make data collection an equal opportunity for all members of society, to improve the planning and implementation of mitigation strategies (Road kill, 2020). The “RoadWatch” app aims to map road kill hotspots and determine those species which are worst affected by AVCs [4]. In the time of writing “*Saving wildlife on India’s roads needs collaborative and not competitive efforts*”, Saxena [4] reports that the website revealed 873 road kill observations. Subsequently this app will use the collected road kill data to assess current mitigation measures. In regard to “RoadWatch”, particularly, mapping of road mortality hotspots bares similarities with GIS applications in spatial and temporal analyses that examine the present clusters. However, the presence of these two apps launched by different organisations of the same country risks skewing data in the different areas visited [4]. A coalesced effort by both organisations would unify findings and better advise mitigation measures.

## 4. Conclusions and Recommendations

This review paper critically analysed the patterns, factors and impacts of AVCs; paying particular attention to the subsequent road kills. A brief overview of the road’s relationship with fauna in the surrounding areas was provided. The development and expansion of road infrastructure was evaluated. How road kills are reported, recorded and observed was identified. Different methods were determined, and their differences, evaluated. The spatiotemporal patterns of road kills were critically examined, focusing on underlying factors that influence AVC clusters. Spatial proximity, road infrastructure, traffic volume and velocity, landscape, climate and weather, and animal behavior, are especially influential over where and when road kills occur. The current mitigation strategies used to address ARAs were then examined and evaluated. Challenges facing road kill mitigation strategies were identified and subsequently analysed. The current GIS tools, applications and other technologies used in road kill studies were outlined and reviewed. These were KDE, hotspot analysis and road kill reporting apps such as “RoadWatch” and “Road kill”. After analysis, this review paper provided recommendations for more holistic road kill studies, increased spatiotemporal coverage of observations, greater usage of citizen science, and the development of road kill apps that better incorporate GIS.

## Figures and Tables

**Figure 1 animals-11-00799-f001:**
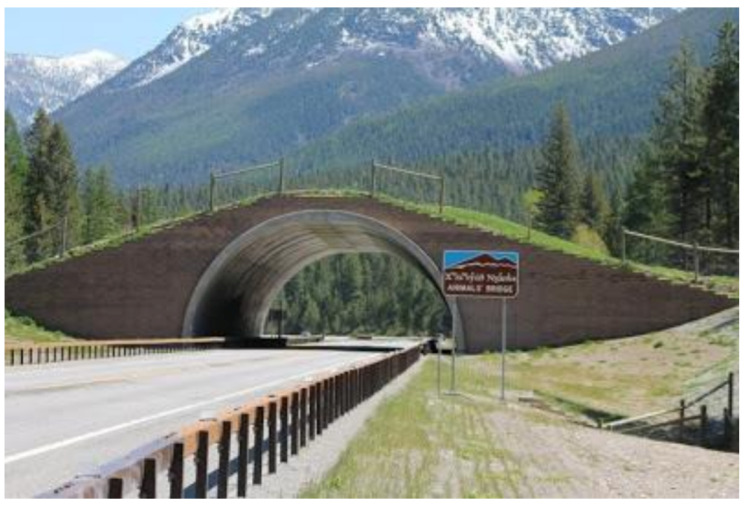
An example of an overpass in Montana, USA.

**Figure 2 animals-11-00799-f002:**
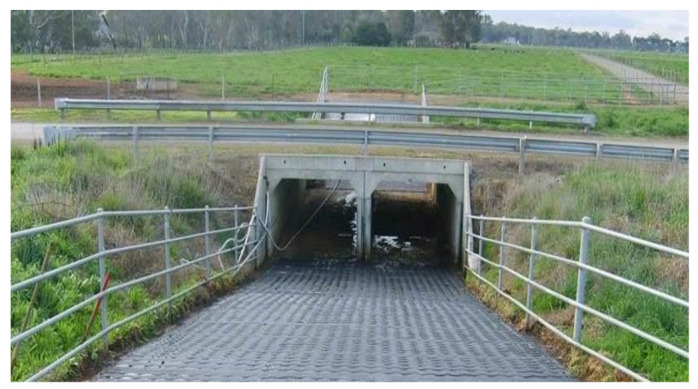
An example of an underpass in Victoria, Australia.

**Figure 3 animals-11-00799-f003:**
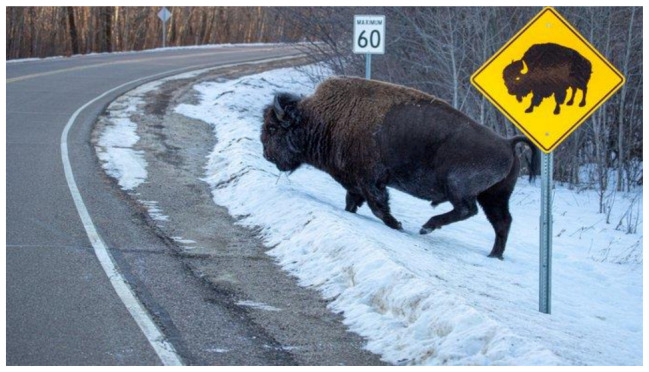
An example of a crossing at the road (name unavailable) surface in the USA.

## Data Availability

Not applicable.

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
