# Peer review of "Spatio-Temporal Patterns and Consequences of Road Kills: A Review"

_animals, 2021, doi:10.3390/ani11030799_

Round 1

Reviewer 1 Report

This paper reviews relevant literature regarding the nature and impacts of animal-related accidents (ARAs), as well as the consequent road kills. The paper aims to provide a cogent evaluation of existing knowledge on the subject matter.

The article is well written and covers all the arguments related to AVCs. It is a complete review and will be of great help to those approaching these issues in the future. It will certainly be cited a lot and will also be very useful for those who have to make generic citations on the topic.

From what I have been able to verify, several sentences could have more citations among the papers viewed. While not affecting the result, considering the probable success that the paper will have, I propose to the author to add a greater number of citations for most of the sentences. However, I would like to underline that, based on my experience, all the sentences are confirmed not only in the literature viewed but also in articles that do not fall within the chosen journals.

Author Response

Dear reviewer,

Reviewer 2 Report

Introduction

There is a great deal of repetition, especially in the Introduction and the length of the paper should be reduced somewhat. Just one example: “…kills typically focuses on a specific roadway within a locale, instead of all the roads in a country…” has been mentioned numerous times.

156      Journal of applied ecology should be Journal of Applied Ecology

554      It is not clear which the bridges are referring to. Are these bridges used by vehicles and the animals cross underneath, or are they used by animals as an overpass?

Fig. 1- 3 All photos are of poor quality, either too dark with no detail or out of focus. Photos must be of high quality.

Appendix A serves no purpose because all the full references are listed anyway below that.

Author Response

Dear reviewer,
